# The reliability and validity of a non-wearable indoor positioning system to assess mobility in older adults: A cross-sectional study

Isabel B. Rodrigues[1,2], Patricia Hewston[1], Jonathan Adachi[1], Sayem Borhan[3,4], George Ioannidis[1], Alexa Kouroukis[1,5], Carolyn Leckie[1], Andrea Lee[6], Alexander Rabinovich[7], Parthipan Siva[8,9], Rachel Swance[10], Suleman Tariq[1,5], Lehana Thabane[3,4,11], Alexandra Papaioannou[1,3]*

1 Department of Medicine, Faculty of Health Sciences, McMaster University, Hamilton, Ontario, Canada, 2 Department of Community Health Sciences, Rady Faculty of Health Sciences, University of Manitoba, Winnipeg, Manitoba, Canada, 3 Department of Health Research Methods, Evidence, and Impact, Faculty of Health Sciences, McMaster University, Hamilton, Ontario, Canada, 4 Biostatistics Unit, St Joseph's Healthcare Hamilton, Hamilton, Ontario, Canada, 5 Faculty of Medicine, University of Toronto, Toronto, Ontario, Canada, 6 Hamilton Health Sciences, Research Development and Innovation, Hamilton, Ontario, Canada 7 Department of Surgery – Division of Orthopaedic Surgery, Faculty of Medicine, McMaster University, Hamilton, Ontario, Canada 8 Chirp Inc., Waterloo, Ontario, Canada 9 Faculty of Engineering, University of Waterloo, Waterloo, Ontario, Canada 10 Faculty of Science, School of Interdisciplinary Science, McMaster University, Hamilton, Ontario, Canada 11 Faculty of Health Science, University of Johannesburg, Johannesburg, South Africa,

* papaioannou@hhsc.ca

## Abstract

### Background

Detecting early changes in walking speed can allow older adults to seek preventative rehabilitation. Currently, there is a lack of consensus on which assessments to use to assess walking speed and how to continuously monitor walking speed outside of the clinic. Chirp is a privacy-preserving radar sensor developed to continuously monitor older adults' safety and mobility without the need for cameras or wearable devices. Our study purpose was to evaluate the inter-sensor reliability, intrasession test-retest reliability, and concurrent validity of Chirp in a clinical setting.

### Methods

We recruited 35 community-dwelling older adults (mean age 75.5 (standard deviation: 6.6) years, 86% female). All participants lived alone in an urban city in southwestern Ontario and had access to a smart device with wireless internet. Data were collected with a 4-meter ProtoKinetics Zeno™ Walkway (pressure sensors) with the Chirp sensor (radar positioning) at the end of the walkway. We assessed participants walking speed during normal and adaptive locomotion experimental conditions (walking-while-talking, obstacle, narrow walking, fast walking). We selected walking speed as a measure as it is a good predictor of functional mobility but also is associated with

**Data availability statement:** See "S3 File. De-identified Data" in the supplementary information

**Funding:** AP received funding from the Federal Economic Development Agency for Southern Ontario (FedDev) -Southern Ontario Pharmaceutical and Health Innovation Ecosystem (SOPHIE) and Chirp Inc.

**Competing interests:** One of the authors of this paper (PS) is a co-owner of Chirp. He reviewed and provided input on the final manuscript. We confirm this does not alter our adherence to PLOS ONE policies on sharing data and materials.

physical and cognitive functioning in older adults. Each of the experimental conditions was conducted twice in a randomized order, with fast walking trials performed last. For intra-session reliability testing, we conducted two blocks of walks within a participant session separated by approximately 30 minutes. Intraclass Correlation Coefficient$_{(A,1)}$ (ICC$_{(A,1)}$) was used to assess the reliability and validity. Linear regression, adjusted for gender, was used to investigate the association between Chirp and cognition and health-related quality of life scores.

## Results

Chirp walking speed inter-sensor reliability ICC$_{(A,1)}$ = 0.999[95% Confidence Interval [CI]: 0.997 to 0.999] and intrasession test-retest reliability [ICC$_{(A,1)}$ = 0.921, 95% CI: 0.725 to 0.969] were excellent across all experimental conditions. Chirp walking speed concurrent validity compared to the ProtoKinetics Zeno™ Walkway was excellent across experimental conditions [ICC$_{(A,1)}$ = 0.993, 95% CI: 0.985 to 0.997]. We found a weak association between walking speed and cognition scores using the Montreal Cognitive Assessment across experimental conditions (estimated β-value = 7.79, 95% CI: 2.79 to 12.80) and no association between walking speed and health-related quality of life using the 12-item Short Form Survey across experimental conditions (estimated β-value = 6.12, 95% CI: -7.12 to 19.36).

## Conclusion

Our results demonstrate that Chirp is a reliable and valid measure to assess walking speed parameters in clinics among older adults.

## Introduction

Mobility impairments can negatively influence quality of life for older adults, especially for those who would like to live independently in their home [1]. Mobility limitations increase the risk of falls [2]. Falls are a major risk for older adults that can result in serious physical and psychological consequences [3,4]. Approximately one third to one half of individuals 65 years of age or older report difficulties with walking or climbing the stairs [5–7]. Mobility impairments, including changes in walking speed, are early indicators of health decline and subsequent disability [3,8]. Early detection and interventions to prevent further limitations could be of benefit for older adults to maintain and regain their daily activity levels and independence [5]. There are significant efforts being made to increase our understanding of and capacity to address mobility impairments and falls among older adults [5,9].

Walking speed and step length can be used to identify older adults at high risk of falls [10,11]. In a clinical setting, healthcare providers use methods based on physical examination and functional tests such as the walking speed test, the Short Physical Performance Battery [12], and the Timed Up and Go Test [13,14] to identify quantitative aspects of walking speed [10]. These functional tests only provide a

snapshot of the patient's walking speed. Additionally, such physical examinations and functional measures take a considerable amount of time to perform in clinic and may not always be feasible to implement under real-world conditions due to limitations in space and access to equipment [10]. Moreover, many first line clinics do not have access to physiotherapy and occupational therapy who evaluate other walking parameters that could predict fall risk including cadence, stride length, or swing phase time [10].

Performing continuous mobility assessments of walking speed may be one solution to help healthcare providers intervene sooner and reduce injuries related to falls and mobility impairments [15,16]. A means of continuously assessing mobility passively during normal activities of daily living allows for real-time mobility assessment and early identification of mobility declines and impairments [17]. Early identification of mobility declines can result in pre-emptive interventions such as physiotherapy or functional strength and balance training to aid or improve mobility [18]. Identifying new technologies in rehabilitation could help identify early declines in mobility as such technologies may be opportunities to monitor health status at home. Systems for at home walking speed analysis can be divided into three major groups: non-wearable systems, wearable systems, and a combination of non-wearable and wearable systems [19]. Each system can provide precise information on walking patterns, step speed, step length and width, and static and dynamic balance under controlled conditions [19]. While there are several precise systems, there are limitations including high costs to purchase the devices, the usability of the system (e.g., requires regular calibration to reduce errors), and the limited number of studies regarding the system's effectiveness [20–22]. Wearable systems such as smartphone apps are a popular method to collect data on gait parameters including gait speed and step length [23]. Although smartphone use is on the rise among older adults, its use is restricted by the need to always carry the smartphone and be connected to Wi-Fi [23]. Other technologies such as virtual reality, augmented reality, and robotics should not be used independently for home-based rehabilitation due to usability and safety reasons [20]. Non-wearable, passive technologies with minimal set-up and calibration may be a more user-friendly solution for older adults [17,19].

Chirp (Waterloo, Ontario, Canada) is a privacy-preserving, non-wearable sensor that uses radar to passively collects walking parameters in older adults [16]. The radar-based approach in the Chirp technology can also be used to monitor walking parameters within the home and in clinic and requires minimal set-up and calibration from the end-user [16]. Since radar technology is safe, easy to use, inexpensive, and inconspicuous while maintaining privacy, it has emerged as the best option for ongoing walking monitoring at home [24]. The Chirp sensor may be a realistic technology to assess mobility in clinical settings. Before the Chirp sensor can be used in clinic, the system's psychometric properties should be tested against standard clinical mobility assessment metrics. This study evaluated the inter-sensor reliability, intrasession test-retest reliability, and concurrent validity of the Chirp sensor among older adults in a clinical setting. Our secondary objective was to determine if there is an association between walking speed using the Chirp sensor and self-reported mobility disability and cognition.

## Methods

### Study design

This study is part of a larger study that also determined the feasibility and acceptability of the Chirp device among community-dwelling older adults. The results of the feasibility and acceptability study are published elsewhere [25]. We conducted a cross-sectional, single-centered study. We followed the STROBE 2007 guidelines for reporting observational studies (S1 Table) [26]. Ethics approval was obtained from the Hamilton Integrated Research Ethics Board (HIREB #15237).

### Setting

We recruited participants from physicians' offices, the local newspaper, and social media. The clinicians (JA, AR, and AP) on our team used a pre-screening referral form to identify potential participants in clinic. Our research team also screened

potential individuals over the telephone to determine eligibility. We recruited participants between February to July 2023. We obtained written informed consent from each participant prior to enrolling them in the study. Participants attended one study visit in a private room at the Hamilton Health Sciences. We provided free transportation for participants with limited mobility or free parking at the clinic. We scheduled participants on a rolling basis.

## Participants

We included participants if they spoke English or attended with a translator or caregiver, were ≥65 years and older, had a Morley Frail Scale score of 3 or more on the FRAIL Scale [27], and were able to follow two-step commands. We excluded individuals who did not live alone, did not have access to Wi-Fi or a smartphone, and required a wheelchair due to medical conditions.

## Data sources/measurement

**Chirp.** The Chirp sensor uses the Texas Instrument IWR6843AOP mmWave radar sensor, a very low resolution (32x32 pixel) thermal sensor, and a microphone to continuously monitor daily activities (transfer times, walking speed, sedentary times, etc.) in the home [16]. The sensor is approved by Innovation, Science and Economic Development Canada (ISED) (IC:29827-CHIRP01T) and Federal Communications Commission (FCC) of the United States of America (FCC ID:2A9Q4-CHIRP01T) for continuous use in indoor environments. For this study, only the 60 to 64 GHz frequency-modulated continuous-wave radar sensor is used to collect a sparse 3D point cloud of moving objects in the scene. The point cloud is captured at 10 frames per second (10 Hz), in the 120-degree field of view of the sensor, up to 8 meters from the sensor. The collected point cloud is analyzed by Chirp's artificial intelligence algorithms to detect and track people and measure their walking speed and the actions they perform (e.g., sit-down, standup, falls, etc.). The captured data can not be used to identify individuals.

The Chirp device is a privacy preserving device that will not collect or store personal identifying information or personal health information. The Chirp device was linked to a unique study identification number for each participant and participants cannot be identified using the Chirp device or smartphone app. For the purposes of this study, the Chirp device did not record or analyze audio, and the industry sponsor ensured these applications remained disabled for the duration of the study.

**Protokinetics Zeno™ Walkway.** We collected data on gait parameters (i.e., gait speed) using a 4-meter ProtoKinetics Zeno™ Walkway (Havertown, PA, USA) at a sampling frequency of 100 Hz. We utilized a standard protocol from the InCIANTI protocol to collect our data [28]. The walkway is a portable, roll-up design that required set up prior to each study visit. To guarantee that the Protokinetics Zeno™ Walkway set up was standardized, we marked the mat's location on the floor with masking tape. We used the ProtoKinetics Movement Analysis Software (PKMAS) to capture our gait parameters.

## Variables

We collected demographic characteristics using an equity lens: Place of residence, Race/ethnicity, Occupation, Gender and sex, Religion, Education, Socioeconomic status, and Social capital (PROGRESS) [29]. Frailty scores were assessed using the Fit-Frailty Assessment and Management Application [30], health-related quality of life using the 12-Item Short Form (SF-12) questionnaire [31], and cognitive status using the Montreal Cognitive Assessment (MoCA) version 8.2 English [32]. Frailty scores were interpreted as follows: not frail- frail scores < 0.24, pre-frail scores 0.18 to 0.24, and frail >0.24 [30]. The SF-12 was scored by yielding two summary measures: the Physical Component Summary and the Mental Component Summary [31]. Scores above 50 indicate a better than average health-related quality of life on the SF-12, while a score below 50 suggest below-average health [31]. The MoCA was scored on a scale of 30 points, where a score

of ≥ 26 points indicate normal cognition, 18 to 25, mild cognition, 10 to 17, moderate, and fewer than 10 points, severe cognitive impairment; MoCA scores were adjusted for education status [32].

## Data collection

The study visit was divided into three blocks. During "Block 1", we assessed gait parameters using the Protokinetics Zeno™ Walkway and the Chirp sensor. During "Block 2", we collected demographic characteristics using the PROG-RESS questionnaire [29], frailty scores with the Fit-Frailty Assessment and Management Application [30], cognitive scores using the MoCA [32], and health-related quality of life with the SF-12 questionnaire [31]. During "Block 3", we repeated the measures of the first block to collect gait parameters for a second time using the Protokinetics Zeno™ Walkway and the Chirp sensor. To determine the intrasession test-retest reliability of the Chirp sensor, the InCIANTI protocol [28] was repeated at the end of the testing session (i.e., Block 3). The Chirp sensors were positioned at the end of the walking path at 2.03 meters from the end of the ProtoKinetics Walkway at a sampling frequency of 10 Hz (Fig 1). The first Chirp sensor (herein known as Chirp 126) was set as the primary sensor, while Chirp 201 was the secondary sensor. The ProtoKinetics Zeno™ Walkway (pressure sensors) and the Chirp sensor (radar positioning) collected data simultaneously. Participants started and ended at the gait mat. Using the InCIANTI protocol [28], participants walked along a 4-meter path during normal and adaptive locomotion experimental conditions: 1) <u>normal walking:</u> self-selected comfortable walking pace, 2) <u>fast walking</u>: walk as fast as possible, 3) <u>obstacle crossing</u>: cross over two obstacles placed in the path while walking as fast as possible, 4) <u>narrow-path</u>: walk at their usual pace but stay between the lines of coloured tape placed 25 cm apart, and 5) <u>walking-while-talking</u>: walking normally while simultaneously performing a verbal cognitive task (i.e., asked to recite names of animals starting with the letter "p"). For the obstacle crossing, the obstacles were 11.4 cm in height and 97.8 cm in length and positioned at 135 cm and 265 cm, respectively, from the starting line of the Protokinetics Zeno™ Walkway. Each experimental condition was conducted twice (during Blocks 1 and 3) in a randomized order with the exception of fast walking trials, which were performed last in each experimental block to avoid any influence on the speed of the preceding trials. To sync the sensor to assess concurrent validity, start and end times for each walkover were defined from the first pressure-mat heel strike to the final pressure-mat toe-off time as defined by ProtoKinetics Movement Analysis Software (PKMAS) [28]. These values were precisely distance-aligned with frames captured by the Chirp sensor, and direct comparisons were made for each walkover. Blocks 1 and 3 were separated by approximately 30 minutes to collect data on intra-session reliability. During each study visit, two to three research assistants were present in the room. Participants were allowed to use assistive devices when performing each walk since the Zeno PKMAS software can identify and remove assistive device tracks to capture an accurate walking speed of each individual.

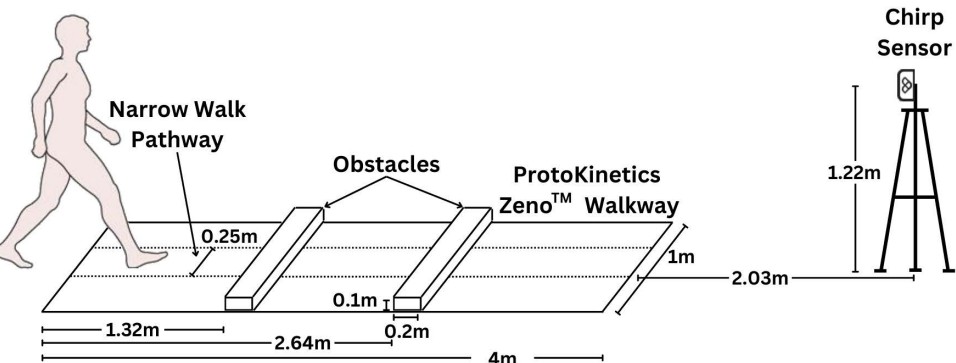

**Fig 1. In-clinic setup of the 4-meter ProtoKinetics Zeno™ Walkway Gait Analysis System and the Chirp sensor for testing concurrent validity of walking speed.** Obstacles were only used for obstacle crossing. Narrow walk pathway is used for narrow walking scenario only.

## Sample size

The approach to assess reliability is based on estimations of the Intraclass correlation (ICC) and the 95% confidence intervals (CI). Thus, we aimed to select a sample size which improves the precision of the estimate. We estimated we would need a sample size of 35 based on the primary outcome (i.e., test-retest reliability) at a 0.05 level of significance (two-tailed), 80% power, and 10% expected drop-out rate is 33 ($p_0$ = 0.75; $p_1$ = 0.9). A higher $p_0$ indicates greater reliability, with $p_0$ = 0.8 indicating the highest acceptable level of reliability [33].

## Outcomes

Our primary outcome was to assess the psychometric properties of the Chirp sensor under laboratory conditions. We assessed two types of reliability: the inter-sensor reliability and intrasession test-retest reliability. Inter-sensor reliability is assessed between Chirp 126 (primary sensor) and Chirp 201 across all experiment conditions and each of the five walks. Similarly, intrasession test-retest reliability was determined by correlating average gait speeds between Block 1 and 3 for all five types of walks (i.e., all walks combined, normal walk, obstacle walk, walk and talk, narrow walk, and fast walk). To assess the concurrent validity of gait speed using the Chirp sensor, comparisons were made against the gold standard utilizing the 4-meter ProtoKinetics Zeno™ Walkway at a sampling frequency of 100 Hz. Concurrent validity was assessed using gait speed measured with the Protokinetics Zeno™ Walkway and Chirp 126 (primary sensor) across all experimental conditions and each of the five walks. Our secondary objective was to determine if there was a correlation between MoCA scores or self-reported mobility disability using the SF-12 and gait speed using Chirp 126.

## Statistical analysis

Descriptive summaries were presented as mean and standard deviation (SD) for continuous variables, and frequency for categorical variables. Intraclass correlation$_{(A,1)}$ (ICC$_{(A,1)}$) was used to assess the concurrent validity of gait speed collected from the Chirp sensor and Protokinetics Zeno™ Walkway, the inter sensor reliability between Chirp 126 and Chirp 201, and the test-retest (Chirp 126, Block 1 versus 3) across all trials for all and individual (normal walking, walking-while-talking, obstacle, narrow walking, fast walking) experimental conditions. The estimated intraclass correlation coefficients, along with 95% confidence intervals (CI), were reported. Linear regression, adjusted for gender, was used to investigate the association between Chirp 126 and MoCA and SF-12. The estimated regression coefficient (β-value) along with 95% CIs were reported. All statistical tests were two-sided with the level of significance of 0.05. All analyses were performed in R version 3.4.2 (R Foundation for Statistical Computing, Vienna, Austria).

## Results

We approached 80 individuals, and after the screening process, we enrolled 35 individuals (Fig 2). Demographic characteristics of participants who chose to join the study are presented in Table 1. Of the 700 walks ([35 participants] × [blocks 1 and 3] × [5 experimental conditions] x [2 trials each]), 47 walks were not collected as six participants did not feel comfortable completing some of the walks; five participants did not complete the narrow walk and/or the obstacle walk, and one participant lost their balance during the first narrow walk and could not complete the subsequent walks including the obstacle crossing. Five of the six participants used a walker. Walking data was collected using Chirp 126, Chirp 201, and the Protokinetics Zeno™ Walkway. Chirp 126 recorded 624/653 walks (95.6%), with 29 missing due to low bandwidth. Chirp 201 recorded 576/653 walks (88.2%), with 24 missing due to low bandwidth and 53 due to a broken sensor. The Protokinetics Zeno™ Walkway recorded 653/653 walks (100%) with no missing data. The results for the walking speed are presented in Table 2.

We conducted an inter-sensor reliability of walking speed between the Chirp sensors 126 and 201. The Chirp walking speed inter-sensor reliability [ICC$_{(A,1)}$ = 0.999 (95% CI: 0.997, 0.999)] and intrasession test-retest reliability [ICC$_{(A,1)}$ =

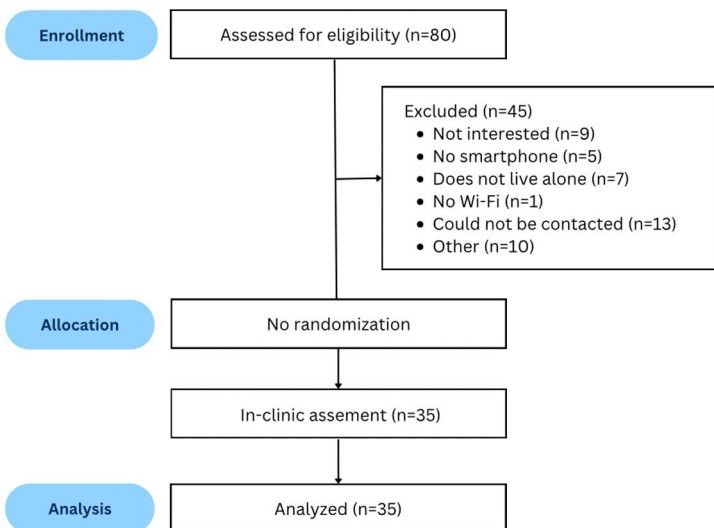

**Fig 2. Participant recruitment and enrollment flow diagram.**

0.921 (95% CI: 0.725,0.969)] were excellent across experimental conditions (see Table 3). A copy of the de-identified data is available in S2 File.

The Chirp walking speed concurrent validity compared to the gold-standard ProtoKinetics Zeno™ Walkway was excellent across experimental conditions [$ICC_{(A,1)}$ = 0.993 (95% CI: 0.985 - 0.997)] (see Table 4).

We found a weak association between walking speed and mean MoCA scores across experimental conditions and no association between walking speed and SF-12 across experimental conditions (see Table 5).

The results of the Bland-Altman analysis can be found in S1 File. The estimated biases were very small; however, in some cases, the estimated biases were significant (significant values are highlighted in yellow in the S1 File). There were no patterns in the Bland-Altman plots.

## Discussion

The Chirp sensor is a privacy-preserving, non-wearable sensor that uses a radar sensor to passively collect gait parameters. The main purpose of our study was to evaluate the inter-sensor reliability, intrasession test-retest reliability, and concurrent validity of Chirp in older adults in an outpatient setting. We recruited 35 community-dwelling older adults who lived alone in a moderate-sized urban city in southwestern Ontario. Data were collected with a 4-meter ProtoKinetics Zeno™ Walkway (pressure sensors) and with the Chirp sensor (radar). Participants walked during normal and adaptive locomotion experimental conditions (i.e., walking-while-talking, obstacle, narrow walking, fast walking). Each of the five experimental conditions was conducted twice in a randomized order, with fast walking trials performed last. Two blocks of walks were conducted within a participant session, separated by approximately 30 minutes, for intra-session reliability testing. We found the Chirp walking speed inter-sensor reliability and intrasession test-retest reliability were excellent across all experimental conditions [34]. The concurrent validity of the Chirp walking speed compared to the gold-standard ProtoKinetics Zeno™ Walkway was also excellent across all experimental conditions. We found a weak association between walking speed and the MoCA across all experimental conditions but no association between Chirp and the SF-12. Our results indicate that this type of radar has excellent validity and reliability to measure gait parameters in older adults in an outpatient setting and could be used to discretely detect subtle gait changes that appear among older adults.

ONE | https://doi.org/10.1371/journal.pone.0307347   April 25, 2025                                                                 7 / 14

**Table 1. Demographic and other health characteristics of participants at baseline (n = 35).**

| Demographic and health characteristics | |
|---|---|
| Age (in years). mean (SD) | 75 (6.6) |
| Height (in cm), median (Q1, Q3) | 165.0 (157.2, 169.1) |
| Weight (in kg), median (Q1, Q3) | 72.6 (59.4, 88.8) |
| BMI, median (Q1, Q3) | 26.6 (23.0, 29.3) |
| Sex, n (%) | |
| Female, n (%) | 30 (86%) |
| Ethnicity, n (%) | |
| Caucasian | 34 (97%) |
| Indigenous | 1 (3) |
| Highest Level of Education, n (%) | |
| Grade school | 1 (3%) |
| High school | 11 (31%) |
| Higher education (college or university) | 23 (65%) |
| Annual income, 2023 CAD | |
| <20,000 | 2 (6%)* |
| 20,001 to 40,000 | 13 (37%)* |
| 40,001 to 60,000 | 11 (31%)* |
| >60,000 | 5 (14%)* |
| Prefer not to answer | 4 (11%)* |
| Place of Residence, n (%) | |
| In the community alone | 35 (100%) |
| Medical history, n (%) | |
| Cancer | 9 (26%) |
| Cardiovascular | 16 (46%) |
| Hearing impairment | 14 (40%) |
| Joint disease | 19 (54%) |
| Respiratory | 2 (6%) |
| Frail Score (using Fit-Frailty Scale), mean (SD) | 0.2 (0.1) |
| Not pre-frail or frail, n (%), mean (SD) | 15 (43%), 0.1 (0.04) |
| Pre-Frail, n (%), mean (SD) | 10 (29%), 0.2 (0.01) |
| Frail, n (%), mean (SD) | 10 (29%), 0.3 (0.08) |
| MoCA Scores, mean (SD) | 23.4 (3.6) |
| Not frail, mean (SD) | 24.5 (3.2) |
| Pre-Frail, mean (SD) | 23.4 (4.6) |
| Frail, mean (SD) | 22.4 (3.8) |

*Note: Rounding error does not add up to 100%.

The present work provides supporting evidence that the Chirp sensors can accurately and reliably assess gait across several different types of walks that older adults may exhibit in an outpatient setting. A brief review of the literature indicates there are few studies that use radar to assess gait parameters in older adults, especially those who may be pre-frail or frail. Nevertheless, of the studies that do exist indicate our validity and reliability results are similar to or better than other studies that used radar to assess gait characteristics in clinical settings. In the Boroomand study, the authors collected gait speeds among 14 healthy young adults using a 24 GHz radar sensor during three different walks (i.e., slow,

**Table 2. Walking speed (meters/second) for the Protokinetics Zeno™ Walkway, Chirp sensor 126, and Chirp sensor 201.**

| Walking condition | Protokinetics Zeno™ Walkway | Chirp sensor 126 | Chirp sensor 201 |
|---|---|---|---|
| Normal walk | 0.974, n = 139 | 0.954, n = 133 | 0.947, n = 122 |
| Obstacle walk | 0.856, n = 116 | 0.854, n = 111 | 0.847, n = 103 |
| Walk and talk | 0.698, n = 139 | 0.694, n = 132 | 0.686, n = 122 |
| Narrow walk | 0.988, n = 123 | 0.971, n = 116 | 0.962, n = 109 |
| Fast walk | 1.254, n = 136 | 1.224, n = 132 | 1.221, n = 120 |

n – number of walks collected

**Table 3. Results of the inter-sensor reliability and test-retest reliability of the Chirp sensors.**

| Chirp 126 (primary) versus Chirp 201 | Inter-sensor reliability ICC (95% CI) |
|---|---|
| All experimental conditions | 0.999 (0.997, 0.999) |
| Normal walk | 0.997 (0.994, 0.999) |
| Obstacle walk | 0.996 (0.990, 0.998) |
| Walk and talk | 0.999 (0.998, 1.000) |
| Narrow walk | 0.996 (0.992, 0.998) |
| Fast walk | 0.994 (0.988, 0.997) |
| **Chirp 126 (Block 1 versus Block 3)** | **Intrasession test-retest reliability ICC (95% CI)** |
| All experimental conditions | 0.921 (0.725, 0.969) |
| Normal walk | 0.862 (0.706, 0.933) |
| Obstacle walk | 0.919 (0.583, 0.973) |
| Walk and talk | 0.844 (0.544, 0.936) |
| Narrow walk | 0.816 (0.403, 0.930) |
| Fast walk | 0.928 (0.858, 0.964) |
| **Chirp 201 (Block 1 versus Block 3)** | **Intrasession test-retest reliability ICC (95% CI)** |
| All experimental conditions | 0.914 (0.701, 0.967) |
| Normal walk | 0.858 (0.699, 0.933) |
| Obstacle walk | 0.899 (0.496, 0.968) |
| Walk and talk | 0.838 (0.453, 0.939) |
| Narrow walk | 0.816 (0.337, 0.935) |
| Fast walk | 0.919 (0.835, 0.961) |

**Table 4. Results of the concurrent validity of the Chirp sensors to the Protokinetics Zeno™ Walkway.**

| Chirp 126 versus Protokinetics Zeno™ Walkway | Concurrent validity, simple bivariate ICC two-way fixed (95% CI) |
|---|---|
| All experimental conditions | 0.993 (0.985, 0.997) |
| Normal walk | 0.994 (0.973, 0.998) |
| Obstacle walk | 0.991 (0.980, 0.996) |
| Walk and talk | 0.990 (0.981, 0.995) |
| Narrow walk | 0.979 (0.952, 0.991) |
| Fast walk | 0.987 (0.948, 0.995) |

**Table 5. Linear regression analysis between walking speed measured by Chirp and MoCA and physical functioning using the SF-12.**

| Chirp 126 and MoCA | Estimated β-value | 95% CI | p-value |
|---|---|---|---|
| All experimental conditions | 7.79 | 2.79, 12.80 | 0.003 |
| Normal walk | 8.31 | 3.15, 13.47 | 0.003 |
| Obstacle walk | 6.99 | 0.37, 13.62 | 0.039 |
| Walk and talk | 4.5 | −0.84, 9.83 | 0.096 |
| Narrow walk | 6.25 | −0.02, 12.51 | 0.051 |
| Fast walk | 5.09 | 0.92, 9.25 | 0.018 |
| **Chirp 126 and SF-12** | **Estimated β-value** | **95% CI** | **p-value** |
| All experimental conditions | 6.12 | −7.12, 19.36 | 0.353 |
| Normal walk | 6.42 | −6.88, 19.73 | 0.332 |
| Obstacle walk | −5.21 | −21.94, 11.52 | 0.527 |
| Walk and talk | 2.6 | −10.34, 15.54 | 0.685 |
| Narrow walk | −5.43 | −21.36, 10.50 | 0.490 |
| Fast walk | 7.88 | −2.34, 18.10 | 0.126 |

normal, and fast) and correlated the results of the radar sensor walk to the GAITRite mat [35]. Accuracy calculation of the radar sensor to the GAITRite mat revealed the fast gait walk had an average validity of 86%, while the average validity for the normal and low speed walks were 81% and 74%, respectively [35]. Similarly, Saho and colleagues also utilized a 24 GHz micro-Doppler radar compared to a 10-meter walkway to assess walking speed in 19 older adults [36]. Saho and colleagues achieved and accuracy of 78.8% using the radar data with a sensitivity of 64.3%, specificity of 81.8%, and precision of 89.5%. Lastly, Wang and colleagues reported an excellent reliability in step time using a 5.8 GHz pulse Doppler radar with ICC of 97% between the radar and a motion capture system under laboratory conditions in 13 healthy young adults [37]. Compared to our study, our results demonstrate similar or better validity and reliability with the Chirp sensor with a concurrent validity of 99.3% (95% CI 98.5% to 99.7%) and reliability of 99.9% (95% CI 99.7% to 99.9%) using a 4-meter range on the ProtoKinetics Zeno™ Walkway. Overall, our research suggest that the Chirp sensors appear to be a promising technology to collect gait speed in older adults in an in clinic-type environment.

Compared to other systems, radar technology has several benefits over other systems. Wireless technology that uses electromagnetic waves (i.e., radar) to continuously assess gait characteristics in an outpatient setting without the involvement of a healthcare practitioner appear feasible due to the radar's safety, simplicity, low cost, lack of contact, and unobtrusiveness while preserving privacy [24]. The advantage of our study was we collected data on several different types of walks that mimic real world conditions and with a diverse group of older adults who were not frail, pre-frail and frail. Detecting changes in gait are particularly important in the context of preventing falls and monitoring older adults' safety and mobility. Lower gait speeds may be associated with early diagnosis of different physical and cognitive diseases, as well as tracking an individual's progress towards recovery following certain therapy sessions [38]. Furthermore, older persons who have slower gaits are more likely to fall, which can have a serious impact on their capacity to live independently [39]. Thus, continuous analysis of gait may detect deviations from normal speed. Our results indicate that the high validity and the reliability of the Chirp sensor to detect different types of gait parameters may be useful in the clinical setting where gait indicators may predict outcomes that are important to older adults such as falls and mobility. The results achieved in this study also pave the way to explore the use of stand-alone radar-based sensors in long hallways for day-to-day long-term monitoring of gait parameters of older adults. The next steps may be to evaluate the Chirp sensors in situations where multiple individuals are walking to increase its applicability to real world scenarios. The possibility of the coexistence of multiple walking individuals is high, especially in the outpatient setting where the hallways are shared with

several individuals including patients and their caregivers, hospital staff, and healthcare providers. Future studies will need to address the problem of tracking and monitoring multiple walking individuals in the same environment using advanced mathematical algorithms that can remove multipath effects or ghosts; ghosts occur when a transmission signal is bounced of another object [40]. Such algorithms may focus on identifying and separating reflected signals that may occur when two or more individuals walk closely together. In contrast to most other radar systems that only acquired walking speed of a single individual, utilizing multipath effect algorithm methods enables the simultaneous extraction of walking speed from numerous closely spaced participants [40]. Future test with Chirp sensors will need to understand how radar ghost due to multipath return can be mitigated.

Chirp sensors are an in-home device that can simply be mounted without the need to modify the walls. Additionally, this system has sensors that monitor activity without cameras or the need to wear equipment or push a button. This passive method of collecting data may be integrated into clinical practice to aid healthcare professional when making medical decisions. The results of the Chirp sensor can be used in clinical practice in two ways. One, healthcare professionals could access reports through the Chirp system that summarize important gait parameters, although, accessing such data may be time consuming for healthcare professionals who have already establish a routine. Alternatively, introducing radar into clinics may be more practical [41]. Careful consideration should be given to ethical and privacy protection when using radar in clinical practice such as how patient data is stored, training health personnel to establish standardized operational procedures, and ensuring system compatibility with other medical devices [41]. One of the primary ethical concerns associated with using any type of radar system in hospitals and clinical practice is patient privacy [41]. Hospitals will need to develop clear policies and procedures to obtain consent. Consent may involve explaining the purpose of the radar system, the potential risks of technology, and how the data will be collected, used, stored, shared and destroyed [41]. There are still several barriers that need to be considered before radar systems can be full adopted and used in hospital and clinical settings. Our study presents with several strengths. We recruited a diverse group of older adults who were not frail, pre-frail, or frail on the Fit-Frailty Application. We also assessed gait parameters against the gold standard ProtoKinetics Zeno™ Walkway and assessed such parameters using several types of walks that older adults would experience under real-world conditions. Despite our strengths, our study has some limitations that should be addressed. We only assessed participants' walking toward the radar (radial direction), while those walking in other directions (e.g., lateral) were not considered in this study. The validity and reliability of the Chirp sensor to assess gait parameters in other directions should be considered in future studies as arbitrary directions may mimic real-world conditions. Lastly, the majority of participants in our study identified as female and from Caucasian descent, and were well educated, so the generalizability to diverse populations may be limited as these populations often experience mistrust and distrust with tracking sensors [42]. Lastly, it is crucial to remember that the suitability of the findings for the intended use of in-home tracking and walking speed assessment from a clinical standpoint depends on a number of variables, including the particular needs of the application, the level of accuracy and precision required, and the constraints of the sensor configuration and data processing methods employed. Although our study's results show that the Chirp sensor has the ability to track walking parameters and identify various walking activities in clinic, more testing and validation would be required to confirm the system's validity and reliability before it could be used in other settings such as long-term care homes where there may be multiple individuals walking in the hallway. It's also critical to consider the sensor setup's limits such as range.

## Conclusion

Our preliminary findings indicate that the Chirp sensor has the capacity to accurately track gait in the outpatient setting, which increases opportunities for its applications in institutional environments. The Chirp walking speed inter-sensor reliability and intrasession test-retest reliability were excellent across experimental conditions. Chirp sensor's concurrent validity compared to the gold-standard ProtoKinetics Zeno™ Walkway was excellent across experimental conditions. We found a weak association between walking speed and cognition scores using the Montreal Cognitive Assessment across

experimental conditions and no association between the walking speed and health-related quality of life using the 12-item Short Form Survey across experimental conditions. These promising results suggest that this type of radar has good potential for the timely detection of discrete and subtle gait changes that appear among older adults.

## Supporting information

**S1 Table. 2007 STROBE Checklist for Cohort Studies** .
(PDF)

**S1 File. Bland-Altman Analysis.**
(PDF)

**S2 File. De-identified Data.**
(XLSX)

## Author contributions

**Conceptualization:** Isabel B. Rodrigues, Patricia Hewston, George Ioannidis, Alexandra Papaioannou.

**Data curation:** Patricia Hewston, Alexa Kouroukis, Rachel Swance, Suleman Tariq.

**Formal analysis:** Sayem Borhan, Lehana Thabane.

**Funding acquisition:** Alexandra Papaioannou.

**Investigation:** Isabel B. Rodrigues, Patricia Hewston, George Ioannidis, Alexa Kouroukis, Rachel Swance, Suleman Tariq.

**Methodology:** Isabel B. Rodrigues, Patricia Hewston, Jonathan Adachi, George Ioannidis, Alexandra Papaioannou.

**Project administration:** Isabel B. Rodrigues, Patricia Hewston, George Ioannidis, Alexandra Papaioannou.

**Resources:** Jonathan Adachi, Carolyn Leckie, Andrea Lee, Alexander Rabinovich, Parthipan Siva, Alexandra Papaioannou.

**Supervision:** Isabel B. Rodrigues, Patricia Hewston, George Ioannidis, Alexandra Papaioannou.

**Validation:** Sayem Borhan, Lehana Thabane.

**Writing – original draft:** Isabel B. Rodrigues.

**Writing – review & editing:** Isabel B. Rodrigues, Patricia Hewston, Jonathan Adachi, Sayem Borhan, George Ioannidis, Alexa Kouroukis, Carolyn Leckie, Andrea Lee, Alexander Rabinovich, Parthipan Siva, Rachel Swance, Suleman Tariq, Lehana Thabane, Alexandra Papaioannou.

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
