## [Decision Letter · Decision Letter 0]

12 Nov 2024

PONE-D-24-26496The reliability and validity of a non-wearable indoor positioning system to assess mobility in older adults: A cross-sectional studyPLOS ONE

Dear Dr. Rodrigues,

Thank you for submitting your manuscript to PLOS ONE. After careful consideration, we feel that it has merit but does not fully meet PLOS ONE’s publication criteria as it currently stands. Therefore, we invite you to submit a revised version of the manuscript that addresses the points raised during the review process.

We look forward to receiving your revised manuscript.

Kind regards,

Yih-Kuen Jan, PhD

Academic Editor

PLOS ONE

Journal requirements:    When submitting your revision, we need you to address these additional requirements. 1. Please ensure that your manuscript meets PLOS ONE's style requirements, including those for file naming. The PLOS ONE style templates can be found at https://journals.plos.org/plosone/s/file?id=wjVg/PLOSOne_formatting_sample_main_body.pdf and https://journals.plos.org/plosone/s/file?id=ba62/PLOSOne_formatting_sample_title_authors_affiliations.pdf 2. We noticed you have some minor occurrence of overlapping text with the following previous publication(s), which needs to be addressed: https://www.mdpi.com/1424-8220/24/4/1056https://academic.oup.com/gerontologist/article/50/4/443/743504?login=false In your revision ensure you cite all your sources (including your own works), and quote or rephrase any duplicated text outside the methods section. Further consideration is dependent on these concerns being addressed. 3. We note that the grant information you provided in the ‘Funding Information’ and ‘Financial Disclosure’ sections do not match.  When you resubmit, please ensure that you provide the correct grant numbers for the awards you received for your study in the ‘Funding Information’ section. 4. Thank you for stating the following financial disclosure:  [We received funding from the Federal Economic Development Agency for Southern Ontario (FedDev) -Southern Ontario Pharmaceutical and Health Innovation Ecosystem (SOPHIE) and Chirp Inc].  Please state what role the funders took in the study.  If the funders had no role, please state: ""The funders had no role in study design, data collection and analysis, decision to publish, or preparation of the manuscript."" If this statement is not correct you must amend it as needed. Please include this amended Role of Funder statement in your cover letter; we will change the online submission form on your behalf. 5. Thank you for stating the following in the Competing Interests section: [One of the authors on this paper (PS) is a co-owner of Chirp. He reviewed and provided input on the final manuscript. ].  Please confirm that this does not alter your adherence to all PLOS ONE policies on sharing data and materials, by including the following statement: ""This does not alter our adherence to  PLOS ONE policies on sharing data and materials.” (as detailed online in our guide for authors http://journals.plos.org/plosone/s/competing-interests).  If there are restrictions on sharing of data and/or materials, please state these. Please note that we cannot proceed with consideration of your article until this information has been declared.  Please include your updated Competing Interests statement in your cover letter; we will change the online submission form on your behalf.

Reviewers' comments:

Reviewer's Responses to Questions

**Comments to the Author**

1. Is the manuscript technically sound, and do the data support the conclusions?

Reviewer #1: Yes

Reviewer #2: Yes

2. Has the statistical analysis been performed appropriately and rigorously? 

Reviewer #1: Yes

Reviewer #2: Yes

3. Have the authors made all data underlying the findings in their manuscript fully available?

Reviewer #1: No

Reviewer #2: Yes

4. Is the manuscript presented in an intelligible fashion and written in standard English?

Reviewer #1: Yes

Reviewer #2: Yes

5. Review Comments to the Author

Reviewer #1: Comments to the author

The manuscript titled " The reliability and validity of a non-wearable indoor positioning system to assess mobility in older adults: A cross-sectional study" presents an interesting research topic that may benefit researchers and health care (measuring gait parameters in older adults).

The authors have well-written elaborately well and were able to clearly communicate their findings.

The authors also explained the content related to evaluating Chirp's inter-sensor reliability, intra-session test-retest reliability, and concurrent validity in a clinical setting.

Moreover, the study involved 35 older adults with an average age of 75.5 years, most of whom were women living alone in an urban area. Participants walked under various experimental conditions, and their movements were recorded using the Chirp sensor and the ProtoKinetics Zeno™ Walkway.

Including, the results show that Chirp demonstrated excellent reliability and validity across all conditions. However, the association between Chirp and cognitive scores was weak, and there was no significant association between Chirp and health-related quality of life. The findings suggest that Chirp is a reliable tool for monitoring older adults' mobility.

There are some areas that require modifications in order to justify grounds for publication within PLOS ONE. Please see below for my specific and generally requested modifications.

Specific Edits:

1. In the Abstract section, the authors evaluated the inter-sensor reliability, intra-session test-retest reliability, and concurrent validity of Chirp in a clinical setting. However, the term “walking speed” was not mentioned in this section.

2. In the Outcomes section, it is unclear whether the 4-meter ProtoKinetics Zeno™ Walkway and the 12-foot ProtoKinetics Zeno™ Walkway refer to the same measurement. The authors should clarify and ensure consistency in the units of measurement used.

3. In the Data Collection section, the authors stated that “The study visit was divided into three blocks.” However, “Block 2” was not mentioned in the study and should be addressed.

4. In the Results section, the phrase “Of the 700 walks (35 participants × 2 blocks × 5 experimental conditions × 2 trials each)” implies that there were 2 blocks. Is this correct?

5. In Tables 2 and 3, the walking speed for each condition should be presented in both tables for consistency.

6. I suggest the authors include a Bland-Altman analysis (including a Bland-Altman plot) to assess the agreement in evaluating Chirp's inter-sensor reliability, intra-session test-retest reliability, and concurrent validity.

Reviewer #2: 1. Author should add more recent studies that can be related to mobility among older adults. need to compare to existing support system to mobility and what is current lack of mobility support system and how this system are much more better and give difference compare to existing system.

2. is there any results related to acceptance from older adults, for example how older adults rate their comfort when they using this system, and perhaps can add features of this system that are practical and user-friendly the older adults.

3. should include ethical considerations approval and how ethics play role when involving this group of respondents?

4. should add more studies and research in this study

6. PLOS authors have the option to publish the peer review history of their article (what does this mean? ). If published, this will include your full peer review and any attached files.

**Do you want your identity to be public for this peer review?** For information about this choice, including consent withdrawal, please see our Privacy Policy .

Reviewer #1: **Yes: ** Teerawat Kamnardsiri

Reviewer #2: No

---

## [Author Response · Author response to Decision Letter 0]

11 Dec 2024

Thank you for reviewing our manuscript entitled “The reliability and validity of a non-wearable indoor positioning system to assess mobility in older adults: A cross-sectional study". We have addressed your comments below. The line and page numbers are associated with the tracked changes version of the manuscript.

Reviewer #1: The manuscript titled " The reliability and validity of a non-wearable indoor positioning system to assess mobility in older adults: A cross-sectional study" presents an interesting research topic that may benefit researchers and health care (measuring gait parameters in older adults). The authors have well-written elaborately well and were able to clearly communicate their findings. The authors also explained the content related to evaluating Chirp's inter-sensor reliability, intra-session test-retest reliability, and concurrent validity in a clinical setting. Moreover, the study involved 35 older adults with an average age of 75.5 years, most of whom were women living alone in an urban area. Participants walked under various experimental conditions, and their movements were recorded using the Chirp sensor and the ProtoKinetics Zeno™ Walkway. Including, the results show that Chirp demonstrated excellent reliability and validity across all conditions. However, the association between Chirp and cognitive scores was weak, and there was no significant association between Chirp and health-related quality of life. The findings suggest that Chirp is a reliable tool for monitoring older adults' mobility.

There are some areas that require modifications in order to justify grounds for publication within PLOS ONE. Please see below for my specific and generally requested modifications.

Specific Edits:

1. In the Abstract section, the authors evaluated the inter-sensor reliability, intra-session test-retest reliability, and concurrent validity of Chirp in a clinical setting. However, the term “walking speed” was not mentioned in this section.

We included the following sentence in the abstract: “We assessed participants walking speed during normal and adaptive locomotion experimental conditions (walking-while-talking, obstacle, narrow walking, fast walking).” (Page 2, Lines 36-37). We have also modified the following sentence in abstract: “Chirp walking speed inter-sensor reliability ICC(A,1)=0.999[95% Confidence Interval [CI]: 0.997 to 0.999] and intrasession test-retest reliability [ICC(A,1) =0.921, 95% CI: 0.725 to 0.969] were excellent across all experimental conditions. Chirp walking speed concurrent validity compared to the ProtoKinetics ZenoTM Walkway was excellent across experimental conditions [ICC(A,1)= 0.993, 95% CI: 0.985 to 0.997].” (Page 2, Lines 44-49) and “We found a weak association between walking speed Chirp and cognition scores using the Montreal Cognitive Assessment across experimental conditions (estimated β-value= 7.79, 95% CI: 2.79 to 12.80) and no association between the walking speed Chirp and health-related quality of life using the 12-item Short Form Survey across experimental conditions (estimated β-value=6.12, 95% CI: -7.12 to 19.36)” (Page 2, Line 49-53). We have additionally modified these results sentences throughout the paper.

2. In the Outcomes section, it is unclear whether the 4-meter ProtoKinetics Zeno™ Walkway and the 12-foot ProtoKinetics Zeno™ Walkway refer to the same measurement. The authors should clarify and ensure consistency in the units of measurement used.

We searched the document and changed “12-foot” to “4-meter” for consistency. The change was made on page 8, line 225 in the outcome section.

3. In the Data Collection section, the authors stated that “The study visit was divided into three blocks.” However, “Block 2” was not mentioned in the study and should be addressed.

Block 2 refers to the time when demographic data was collected. We have clarified this point with the following sentence: “During “Block 2”, we collected demographic characteristics using the PROGRESS questionnaire [29], frailty scores with the Fit-Frailty Assessment and Management Application [30], cognitive scores using the MoCA [32], and health-related quality of life with the SF-12 questionnaire [31].” (Page 7, Lines 177-178).

4. In the Results section, the phrase “Of the 700 walks (35 participants × 2 blocks × 5 experimental conditions × 2 trials each)” implies that there were 2 blocks. Is this correct?

There were three blocks, so we have rewritten the sentence to clarify this: “Of the 700 walks ([35 participants] × [blocks 1 and 3] × [5 experimental conditions] x [2 trials each])” (Page 9, line 246).

5. In Tables 2 and 3, the walking speed for each condition should be presented in both tables for consistency.

We included the gait speed for each condition in table 2 (see pages 10-11, line 262).

6. I suggest the authors include a Bland-Altman analysis (including a Bland-Altman plot) to assess the agreement in evaluating Chirp's inter-sensor reliability, intra-session test-retest reliability, and concurrent validity.

We included Bland Altman analysis and plot as a supplementary file (see S1 File) and made note of the file in the results section (“The results of the Bland-Altman analysis can be found in S1 File. The estimated biases were very small; however, in some cases, the estimated biases were significant (significant values are highlighted in yellow in the S1 File). There were no patterns in the Bland-Altman plots.” Page 12, Lines 277-280).

Reviewer #2:

1. Author should add more recent studies that can be related to mobility among older adults. need to compare to existing support system to mobility and what is current lack of mobility support system and how this system are much more better and give difference compare to existing system.

We included several studies on different types of devices that can be used to assess mobility in adults and older adults with comorbidities. Of the articles we cited, two are reviews (one scoping review and one literature review, citations 24 and 26). To address your point, we have included the following paragraph in the introduction: “A recent scoping review of 95 reviews (51 systematic reviews and 44 reviews) identified several types of technologies that can be used for in-home rehabilitation [24]. The authors of the scoping review found that technologies such as virtual reality, augmented reality, and robotics should not be used independently for home-based rehabilitation due to usability and safety reasons [24]. Nevertheless, one major finding revealed technologies such as gamification, digital mobile apps, and Internet of Things can increase end-user engagement in the rehabilitation process as it allows for self-monitoring [24]. The biggest limitations to the current devices identified in this scoping review were the high costs to purchase the devices, the usability of the system (e.g., requires regular calibration to reduce errors), and the limited number of studies regarding the systems effectiveness [24–26]. Non-wearable, passive technologies with minimal set-up and calibration may be a more user-friendly solution for older adults [21,23].” (Page 4, lines 87-100).

2. is there any results related to acceptance from older adults, for example how older adults rate their comfort when they using this system, and perhaps can add features of this system that are practical and user-friendly the older adults.

We also conducted a feasibility and acceptability trial with the Chirp device. The results indicate that the Chirp device is a feasible and reliable artificial intelligence technology that can be used at home by community-dwelling older adults. We have referenced the following paper throughout the manuscript (see reference 28):

“A Papaioannou, P Siva, G Ioannidis, IB Rodrigues, J Adachi, A Rabinovich, R Swance, S Tariq, A Kouroukis, C Leckie, A Lee. P Hewston. Chirp artificial intelligence at home. Poster presented at: Canadian Geriatrics Society 2024 Annual Scientific Meeting; Apr 25, 2024. Vancouver, BC.”

3. should include ethical considerations approval and how ethics play role when involving this group of respondents?

We have added the following sentences to address your point: “The Chirp device is a privacy preserving device that will not collect or store personal identifying information or personal health information. The Chirp device was linked to a unique study identification number for each participant and participants cannot be identified using the Chirp device or smartphone app. For the purposes of this study, the Chirp device did not record or analyze audio, and the industry sponsor ensured these applications remained disabled for the duration of the study.” (Page 6, lines 147-152)

4. should add more studies and research in this study

Thank you for your comment. We have addressed the issue under your first comment.

Journal requirements: When submitting your revision, we need you to address these additional requirements.

We have reviewed and confirm that the manuscript meets the PLOS ONE style requirements.

https://www.mdpi.com/1424-8220/24/4/1056

There is some overlap with the above paper as the Chirp study is part of a larger study. The MDPI paper is also part of the larger Chirp stud and utilized the same methods reported in our paper. We have attempted to reword some of the sentences, but there will be slight overlap as the methods are the same between both papers.

In your revision ensure you cite all your sources (including your own works), and quote or rephrase any duplicated text outside the methods section. Further consideration is dependent on these concerns being addressed.

We have updated the Funding Information and Financial Disclosure section: “We received funding from the Federal Economic Development Agency for Southern Ontario (FedDev), the Southern Ontario Pharmaceutical and Health Innovation Ecosystem (SOPHIE), and Chirp Inc. The funders had no role in study design, data collection and analysis, or decision to publish. The funder did play a role in the preparation of the manuscript”.

[We received funding from the Federal Economic Development Agency for Southern Ontario (FedDev) -

Southern Ontario Pharmaceutical and Health Innovation Ecosystem (SOPHIE) and Chirp Inc].

We have modified the statement to the following: We have updated the Funding Information and Financial Disclosure section: “We received funding from the Federal Economic Development Agency for Southern Ontario (FedDev), the Southern Ontario Pharmaceutical and Health Innovation Ecosystem (SOPHIE), and Chirp Inc. The funders had no role in study design, data collection and analysis, or decision to publish. The funder did play a role in the preparation of the manuscript

5. Thank you for stating the following in the Competing Interests section: [One of the authors on this paper (PS) is a co-owner of Chirp. He reviewed and provided input on the final manuscript. ]. Please confirm that this does not alter your adherence to all PLOS ONE policies on sharing data and materials, by including the following statement: ""This does not alter our adherence to PLOS ONE policies on sharing data and materials.” (as detailed online in our guide for authors http://journals.plos.org/plosone/s/competing-interests). If there are restrictions on sharing of data and/or materials, please state these. Please note that we cannot proceed with consideration of your article until this information has been declared.

One of the authors on this paper (PS) is a co-owner of Chirp. He reviewed and provided input on the final manuscript. We confirm that this does not alter our adherence to PLOS ONE policies on sharing data and materials.

We have reviewed our reference list and confirm that it is complete and correct.

---

## [Decision Letter · Decision Letter 1]

5 Jan 2025

PONE-D-24-26496R1The reliability and validity of a non-wearable indoor positioning system to assess mobility in older adults: A cross-sectional studyPLOS ONE

Dear Dr. Rodrigues,

Thank you for submitting your manuscript to PLOS ONE. After careful consideration, we feel that it has merit but does not fully meet PLOS ONE’s publication criteria as it currently stands. Therefore, we invite you to submit a revised version of the manuscript that addresses the points raised during the review process.

We look forward to receiving your revised manuscript.

Kind regards,

Yih-Kuen Jan, PhD

Academic Editor

PLOS ONE

**Journal Requirements:**

Reviewers' comments:

Reviewer's Responses to Questions

**Comments to the Author**

1. If the authors have adequately addressed your comments raised in a previous round of review and you feel that this manuscript is now acceptable for publication, you may indicate that here to bypass the “Comments to the Author” section, enter your conflict of interest statement in the “Confidential to Editor” section, and submit your "Accept" recommendation.

Reviewer #2: All comments have been addressed

2. Is the manuscript technically sound, and do the data support the conclusions?

Reviewer #2: Yes

3. Has the statistical analysis been performed appropriately and rigorously? 

Reviewer #2: Yes

4. Have the authors made all data underlying the findings in their manuscript fully available?

Reviewer #2: Yes

5. Is the manuscript presented in an intelligible fashion and written in standard English?

Reviewer #2: Yes

6. Review Comments to the Author

**Reviewer #2:**  Abstract Enhancement:

1) Consider explicitly emphasizing the novel contributions of the study in the abstract. While the findings are robust, a clearer statement of the implications for clinical applications could enhance the abstract’s impact.

2) The use of "walking speed" could be further clarified to highlight its critical role in mobility assessment.

Introduction:

1) The introduction could benefit from a stronger emphasis on the gaps in current gait assessment technologies and how the Chirp sensor addresses these gaps.

2) Including more recent and diverse studies in the literature review would strengthen the relevance of the research context.

Discussion and Implications:

1) Strengthen the discussion around how this technology could be integrated into existing clinical workflows and what barriers might exist for widespread adoption.

2) Discuss the potential for real-world application in multi-user settings, as highlighted briefly, but elaborate on the challenges and next steps for such scenarios.

7. PLOS authors have the option to publish the peer review history of their article (what does this mean? ). If published, this will include your full peer review and any attached files.

**Do you want your identity to be public for this peer review?** For information about this choice, including consent withdrawal, please see our Privacy Policy .

Reviewer #2: No

---

## [Author Response · Author response to Decision Letter 1]

19 Feb 2025

Thank you for reviewing our manuscript entitled “The reliability and validity of a non-wearable indoor positioning system to assess mobility in older adults: A cross-sectional study". We have addressed your comments below. The line and page numbers are associated with the tracked changes version of the manuscript.

Reviewers' comments:

Reviewer #2: Abstract Enhancement:

Abstract:

1) Consider explicitly emphasizing the novel contributions of the study in the abstract. While the findings are robust, a clearer statement of the implications for clinical applications could enhance the abstract’s impact.

We have included the following statement in the abstract: “Detecting early sign of walking speed decline can allow older adults to seek preventative rehabilitation. Currently, there is a lack of consensus on which assessments to use to assess walking speed and how to continuously monitor walking speed outside of the clinic. Chirp is a privacy-preserving radar sensor developed to continuously monitor older adults' safety and mobility without the need for cameras or wearable devices. Our study purpose was to evaluate the inter-sensor reliability, intrasession test-retest reliability, and concurrent validity of Chirp in a clinical setting.” (Page 2, lines 29 to 35).

2) The use of "walking speed" could be further clarified to highlight its critical role in mobility assessment.

We have included the following statement: “We selected walking speed as a measure as it is a good predictor of functional mobility but also is associated with physical and cognitive functioning in older adults.” (Page 2, lines 41 to 43).

Introduction:

1) The introduction could benefit from a stronger emphasis on the gaps in current gait assessment technologies and how the Chirp sensor addresses these gaps.

We have modified the introduction to emphasis the need to conduct this study using Chirp. Our main changes are in paragraphs 2 and 3: “Walking speed and step length can be used to identify older adults at high risk of falls [14,15]. In a clinical setting, healthcare providers use methods based on physical examination and functional tests such as the walking speed test, the Short Physical Performance Battery [16], and the Timed Up and Go Test [17,18] to identify quantitative aspects of walking speed [14]. These functional tests only provide a snapshot of the patient’s walking speed [14]. Additionally, such physical examinations and functional measures take a considerable amount of time to perform in clinic and may not always be feasible to implement under real-world conditions due to limitations in space and access to equipment [14]. Moreover, many first line clinics do not have access to physiotherapy and occupational therapy who evaluate other walking parameters that could predict fall risk including cadence, stride length, or swing phase time [14].

Performing continuous mobility assessments of walking speed may be one solution to help healthcare providers intervene sooner and reduce injuries related to falls and mobility impairments [19,20]. A means of continuously assessing mobility passively during normal activities of daily living allows for real-time mobility assessment and early identification of mobility declines and impairments [21]. Early identification of mobility declines can result in pre-emptive interventions such as physiotherapy or functional strength and balance training to aid or improve mobility [22]. Identifying new digital technologies in rehabilitation could help identify early declines in mobility as such technologies may be opportunities to monitor health status at home. Systems for at home walking speed analysis can be divided into three major groups: non-wearable systems, wearable systems, and a combination of non-wearable and wearable systems [23]. Each system can provide precise information on walking patterns, step speed, step length and width, and static and dynamic balance under controlled conditions [23]. While there are several precise systems, there are limitations including high costs to purchase the devices, the usability of the system (e.g., requires regular calibration to reduce errors), and the limited number of studies regarding the system’s effectiveness [24–26]. Wearable systems such as smartphone apps are a popular method to collect data on gait parameters including gait speed and step length [27]. Although smartphone use is on the rise among older adults, its use is restricted by the need to always carry the smartphone and be connected to Wi-Fi. Other technologies such as virtual reality, augmented reality, and robotics should not be used independently for home-based rehabilitation due to usability and safety reasons [24]. Non-wearable, passive technologies with minimal set-up and calibration may be a more user-friendly solution for older adults [21,23].

Chirp (Waterloo, Ontario, Canada) is a privacy-preserving, non-wearable sensor that uses radar to passively collects walking parameters in older adults [20]. The radar-based approach in the Chirp technology can also be used to monitor walking parameters within the home and requires minimal set-up and calibration from the end-user [20]. Since radar technology is safe, easy to use, inexpensive, and inconspicuous while maintaining privacy, it has emerged as the best option for ongoing walking monitoring at home [28]. The Chirp sensor may be a realistic technology to assess mobility in clinical settings. Before the Chirp sensor can be used in clinic, the system’s psychometric properties should be tested against standard clinical mobility assessment metrics. This study evaluated the inter-sensor reliability, intrasession test-retest reliability, and concurrent validity of the Chirp sensor among older adults in a clinical setting. Our secondary objective was to determine if there is an association between walking speed using the Chirp sensor and self-reported mobility disability and cognition.” (Pages 4 and 5 lines 83 to 122)

2) Including more recent and diverse studies in the literature review would strengthen the relevance of the research context.

We included several types of reviews to support the introduction; we limited our search to reviews from the last 5-years. Please see the following references:

23. Argañarás JG, Wong YT, Begg R, Karmakar NC. State-of-the-art wearable sensors and possibilities for radar in fall prevention. Vol. 21, Sensors. MDPI; 2021.

24. Arntz A, Weber F, Handgraaf M, Lällä K, Korniloff K, Murtonen KP, et al. Technologies in Home-Based Digital Rehabilitation: Scoping Review. JMIR Rehabil Assist Technol. JMIR Publications Inc.; 2023;10.

25. Do Nascimento LMS, Bonfati LV, Freitas MLB, Mendes Junior JJA, Siqueira HV, Stevan SL. Sensors and systems for physical rehabilitation and health monitoring—a review. Vol. 20, Sensors (Switzerland). MDPI AG; 2020. p. 1–28.

26. Cogollor JM, Rojo-Lacal J, Hermsdörfer J, Ferre M, Teresa Arredondo Waldmeyer M, Giachritsis C, et al. Evolution of cognitive rehabilitation after stroke from traditional techniques to smart and personalized home-based information and communication technology systems: Literature review. Vol. 20, JMIR Rehabilitation and Assistive Technologies. 2018.

27. Brognara L. Gait Assessment Using Smartphone Applications in Older Adults: A Scoping Review. Vol. 9, Geriatrics (Switzerland). Multidisciplinary Digital Publishing Institute (MDPI); 2024.

28. Zeng X, Báruson HSL, Sundvall A. Walking Step Monitoring with a Millimeter-Wave Radar in Real-Life Environment for Disease and Fall Prevention for the Elderly. Sensors. MDPI; 2022 Dec 1;22(24).

Discussion and Implications:

1) Strengthen the discussion around how this technology could be integrated into existing clinical workflows and what barriers might exist for widespread adoption.

We included the following paragraph to discuss how the technology can be integrated into clinical workflow; however, we will need several studies to troubleshoot the ethical implications of implementing such technology in hospitals and clinical practice: “Chirp sensors are an in-home device that can simply be mounted without the need to modify the walls. Additionally, this system has sensors that monitor activity without cameras or the need to wear equipment or push a button. This passive method of collecting data may be integrated into clinical practice to aid healthcare professional when making medical decisions. The results of the Chirp sensor can be used in clinical practice in two ways. One, healthcare professionals could access reports through the Chirp system that summarize important gait parameters, although, accessing such data may be time consuming for healthcare professionals who have already establish a routine. Alternatively, introducing radar into clinics may be more practical [45]. Careful consideration should be given to ethical and privacy protection when using radar in clinical practice such as how patient data is stored, training health personnel to establish standardized operational procedures, and ensuring system compatibility with other medical devices [45]. One of the primary ethical concerns associated with using any type of radar system in hospitals and clinical practice is patient privacy [45]. Hospitals will need to develop clear policies and procedures to obtain consent. Consent may involve explaining the purpose of the radar system, the potential risks of technology, and how the data will be collected, used, stored, shared and destroyed [45]. There are still several barriers that need to be considered before radar systems can be full adopted and used in hospital and clinical settings.” (Page 16, lines 361 to 377).

2) Discuss the potential for real-world application in multi-user settings, as highlighted briefly, but elaborate on the challenges and next steps for such scenarios.

Although the discussion regarding multiple walking individuals is beyond the scope of this paper, we have included the following paragraph to address your point as it is important to mention our next steps: “The next steps may be to evaluate the Chirp sensors in situations where multiple individuals are walking to increase its applicability to real world scenarios. The possibility of the coexistence of multiple walking individuals is high, especially in the outpatient setting where the hallways are shared with several individuals including patients and their caregivers, hospital staff, and healthcare providers. Future studies will need to address the problem of tracking and monitoring multiple walking individuals in the same environment using advanced mathematical algorithms that can remove multipath effects or ghosts; ghosts occur when a transmission signal is bounced of another object [44]. Such algorithms may focus on identifying and separating reflected signals that may occur when two or more individuals walk closely together. In contrast to most other radar systems that only acquired walking speed of a single individual, utilizing multipath effect algorithm methods enables the simultaneous extraction of walking speed from numerous closely spaced participants [44]. Future test with Chirp sensors will need to understand how radar ghost due to multipath return can be mitigated.” (Page 15 and 16, lines 348 to 360).

---

## [Decision Letter · Decision Letter 2]

21 Feb 2025

The reliability and validity of a non-wearable indoor positioning system to assess mobility in older adults: A cross-sectional study

PONE-D-24-26496R2

Dear Dr. Rodrigues,

We’re pleased to inform you that your manuscript has been judged scientifically suitable for publication and will be formally accepted for publication once it meets all outstanding technical requirements.

Kind regards,

Yih-Kuen Jan, PhD

Academic Editor

PLOS ONE

Additional Editor Comments (optional):

Reviewers' comments:

Reviewer's Responses to Questions

**Comments to the Author**

1. If the authors have adequately addressed your comments raised in a previous round of review and you feel that this manuscript is now acceptable for publication, you may indicate that here to bypass the “Comments to the Author” section, enter your conflict of interest statement in the “Confidential to Editor” section, and submit your "Accept" recommendation.

Reviewer #2: All comments have been addressed

2. Is the manuscript technically sound, and do the data support the conclusions?

Reviewer #2: Yes

3. Has the statistical analysis been performed appropriately and rigorously? 

Reviewer #2: Yes

4. Have the authors made all data underlying the findings in their manuscript fully available?

Reviewer #2: Yes

5. Is the manuscript presented in an intelligible fashion and written in standard English?

Reviewer #2: Yes

6. Review Comments to the Author

Reviewer #2: The study mentions ethics approval from the Hamilton Integrated Research Ethics Board (HIREB #15237). Ensure that this information is consistently and prominently stated in both the Methods and any ethics sections to highlight compliance.

Written informed consent was obtained, but consider clarifying any measures taken to protect participants' anonymity, especially in terms of how raw data from sensors was stored and accessed.

Authors have stated that all data are fully available without restriction. Confirm that any potential limitations (e.g., participant privacy or proprietary considerations) have been thoroughly addressed, and that the data-sharing policy aligns with PLOS ONE's requirements.

7. PLOS authors have the option to publish the peer review history of their article (what does this mean? ). If published, this will include your full peer review and any attached files.

**Do you want your identity to be public for this peer review?** For information about this choice, including consent withdrawal, please see our Privacy Policy .

Reviewer #2: No

---

## [Editor Report · Acceptance letter]

PONE-D-24-26496R2

PLOS ONE

Dear Dr. Rodrigues,

I'm pleased to inform you that your manuscript has been deemed suitable for publication in PLOS ONE. Congratulations! Your manuscript is now being handed over to our production team.

Kind regards,

on behalf of

Dr. Yih-Kuen Jan

Academic Editor

PLOS ONE